# Bimodal seismicity in the Himalaya controlled by fault friction and geometry

Luca Dal Zilio [1], Ylona van Dinther [2,3], Taras Gerya [1] & Jean-Philippe Avouac [4]

There is increasing evidence that the Himalayan seismicity can be bimodal: blind earthquakes (up to Mw ~ 7.8) tend to cluster in the downdip part of the seismogenic zone, whereas infrequent great earthquakes (Mw 8+) propagate up to the Himalayan frontal thrust. To explore the causes of this bimodal seismicity, we developed a two-dimensional, seismic cycle model of the Nepal Himalaya. Our visco-elasto-plastic simulations reproduce important features of the earthquake cycle, including interseismic strain and a bimodal seismicity pattern. Bimodal seismicity emerges as a result of relatively higher friction and a non-planar geometry of the Main Himalayan Thrust fault. This introduces a region of large strength excess that can only be activated once enough stress is transferred upwards by blind earthquakes. This supports the view that most segments of the Himalaya might produce complete ruptures significantly larger than the 2015 Mw 7.8 Gorkha earthquake, which should be accounted for in future seismic hazard assessments.

[1] Geophysical Fluid Dynamics, Institute of Geophysics, ETH Zürich, Sonneggstrasse 5, 8092 Zürich, Switzerland. [2] Seismology and Wave Physics, Institute of Geophysics, ETH Zürich, Sonneggstrasse 5, 8092 Zürich, Switzerland. [3] Department of Earth Sciences, Utrecht University, Budapestlaan 6, Utrecht 3584 CD, The Netherlands. [4] Geological and Planetary Sciences, California Institute of Technology, Pasadena, CA 91125, USA. Correspondence and requests for materials should be addressed to L.D.Z. (email: luca.dalzilio@erdw.ethz.ch)

On 25 April 2015, an earthquake with moment magnitude Mw 7.8 struck the Nepal Himalaya[1–3], rupturing a 50-km-wide segment of the Main Himalayan Thrust (MHT) fault (Fig. 1a). The 2015 Gorkha earthquake has a similar location as the 1833 earthquake, with estimated magnitude Mw 7.6–7.7, which also caused significant damage in Kathmandu[4,5]. The geometry of the MHT is relatively well known in their hypocentral region from various geological and geophysical campaigns[6–8]. In particular, geodetic data (SAR, InSAR and GPS) and the detailed location of the Gorkha seismic sequence have provided new constraints on the geometry of the MHT[9,10]. This information allows us to investigate the relation between interseismic strain and seismicity—given the MHT geometry—and contribute to an ongoing debate on how the Himalayan wedge is deforming. Some authors claim that the location of the front of the high topography could be explained by a mid-crustal ramp along the MHT[11–13]. Conversely, others have argued for active out-of-sequence thrusting at the front of the high Himalaya[14,15]. Understanding how and where stresses build up in the Himalaya is important, because evaluating the balance between the interseismic strain accumulation and the elastic strain released during seismic events could potentially improve the seismic hazard assessment in central Nepal following the 2015 earthquake[16].

It has long been noticed that the seismicity in the Himalaya is bimodal[11,17,18]. Partial (blind) earthquakes (up to Mw ∼ 7.8) tend to cluster and repeatedly rupture the deeper portion of the MHT, whereas sporadic great earthquakes (Mw > 8) completely unzip the entire width of the seismogenic zone (Fig. 1a). The partial ruptures are generally characterised by 10–15 km focal depths and clustered along the front of the Himalaya. They seem to occur in the vicinity of the mid-crustal ramp[11]. The Mw 7.8 Gorkha earthquake is the largest known event in that category. On the other hand, paleoseismological field studies found evidence for surface ruptures at the Himalayan frontal fault (Main Frontal Thrust, MFT), probably associated with great (Mw > 8) events[16,19–22]. The 1934 Mw 8.4 Bihar Nepal[16,22] and the 1950 Mw 8.7 Assam earthquake[23]—the largest intracontinental earthquake ever recorded—probably fall in that category. On the basis of these observations, the mechanism driving bimodal behaviour remains poorly understood. One potential explanation is that the MHT consists of along-dip subsegments that rupture—either independently or jointly with neighbouring segments during larger earthquakes—with a non-periodic or even chaotic behaviour arising from stress transfers. This segmentation may partly be controlled by rheological[24] and geometrical complexities such as local non-planarity[5,25,26]. There is also growing evidence that fault frictional properties are also an influential and perhaps determining factor that affect the spatial extent, size and timing of megathrust ruptures[27]. Dynamic simulations over multiple earthquake cycles with a linear slip-weakening friction law show that a large event that ruptures the entire fault is preceded by a number of small events with various rupture lengths[28]. These results are in keeping with dynamic modelling of the seismic cycle based on rate-and-state friction, which produce partial ruptures even in the case of a planar fault with uniform frictional properties[29]. However, how complete ruptures relate to partial ruptures and the geometry and mechanical properties of the MHT, has not yet been investigated quantitatively.

Here we report the use of a novel two-dimensional (2D) numerical approach (Methods section) to explore the seismic rupture pattern on the MHT over many earthquake cycles (Fig. 1b). The geometry and mechanical properties of our model are defined based on constraints from structural geology and geophysical campaigns[7] and new insights gained from studies of the Gorkha sequence[1,5,9,10]. The temperature distribution is based on a thermokinematic model derived from thermochronological and thermobarometric data[13] (Supplementary Fig. 1). The model is kinematically driven using a boundary condition that translates into a convergence rate of 38 mm year$^{-1}$ across the collisional system. The reference geometry of the MHT (Fig. 1b) is inferred from Elliott et al.[9] and denoted as model EF. It is comprised of three segments to reflect the ramp-flat-ramp geometry: a shallow ∼30° dipping ramp between the surface and 5-km depth constrained by structural sections; a flat portion with a shallow angle reaching, finally, a steeper mid-crustal ramp[30]. Uncertainties regarding the geometry of the MHT still exist, and relatively gentle variation in geometry have also been proposed[10]. We therefore also perform numerical experiments considering this alternative, smoother fault model (model DF; Fig. 1c and Supplementary Fig. 2). To test the sensitivity of the model to the fault geometry, we consider a simple planar fault as well (model PF; Fig. 1c and Supplementary Fig. 2). For each of the three fault geometries adopted, we execute a parameter study of the fault frictional properties by testing values of effective static fault friction (that is, static friction ($\mu_s$) including pore-fluid pressure: $\mu_{eff} = \mu_s (1 - \lambda)$) between 0.06 and 0.2 (Supplementary Table 1). This range is consistent with the results of a compilation of previously published data[31]. A detailed description of the numerical technique, model setup, modelling procedure and limitations is given in the Methods section.

## Results

**Consistency with interseismic deformation.** An important goal in Himalayan studies over the past decades has been to refine the Himalayan convergence rate[32,33], because this is responsible for the productivity of Himalayan earthquakes[31,34]. We emulate the observed velocity field by imposing a convergence rate of 38 mm year$^{-1}$. The model produces about 19–20 mm year$^{-1}$ of convergence across the Himalaya, a value consistent with the long-term geological rate, while the residual convergence rate is dissipated by deformation distributed outside the domain shown in Fig. 1b. The model fits the geodetic measurements of interseismic strain remarkably well (Fig. 2a). All three fault geometries yield predictions in good agreement with uplift rates measured from spirit-levelling[35], inSAR[36], and horizontal velocity measured from GPS[8] (Supplementary Fig. 3). However, we note that model EF agrees particularly well with the data, in terms of both horizontal and vertical velocities.

The mid-crustal ramp operates as a geometric asperity during interseismic periods where elastic strain builds up and accounts for as much as two-thirds of the convergence rate (Fig. 2b). At greater depth, the higher temperature favours the transition from frictionally unstable velocity-weakening behaviour to stable (velocity-strengthening) visco-plastic creep (Supplementary Fig. 4). Visco-plastic strain rates show a sub-horizontal shear zone in the middle-lower crust, which corresponds to the aseismic creep along the MHT. Distributed viscous deformation also occurs in the vicinity of the kink along the MHT ramp-flat geometry (Fig. 2b). Another constraint on the simulated tectonic deformation comes from the off-megathrust events. The model shows that anelastic strain off the MHT tends to cluster beneath the topographic front of the Higher Himalaya (Fig. 2c, d). In fact, most of these events concentrate in a narrow zone near the edge of the mid-crustal ramp, which correlates well with the microseismicity observed over the past decades[11]. This off-megathrust earthquake activity also shows a cutoff beneath the Higher Himalaya, which corresponds to the region where the viscous deformation is dominant and the axes of principal compressional stresses ($\sigma_1$) become (sub-)vertical.

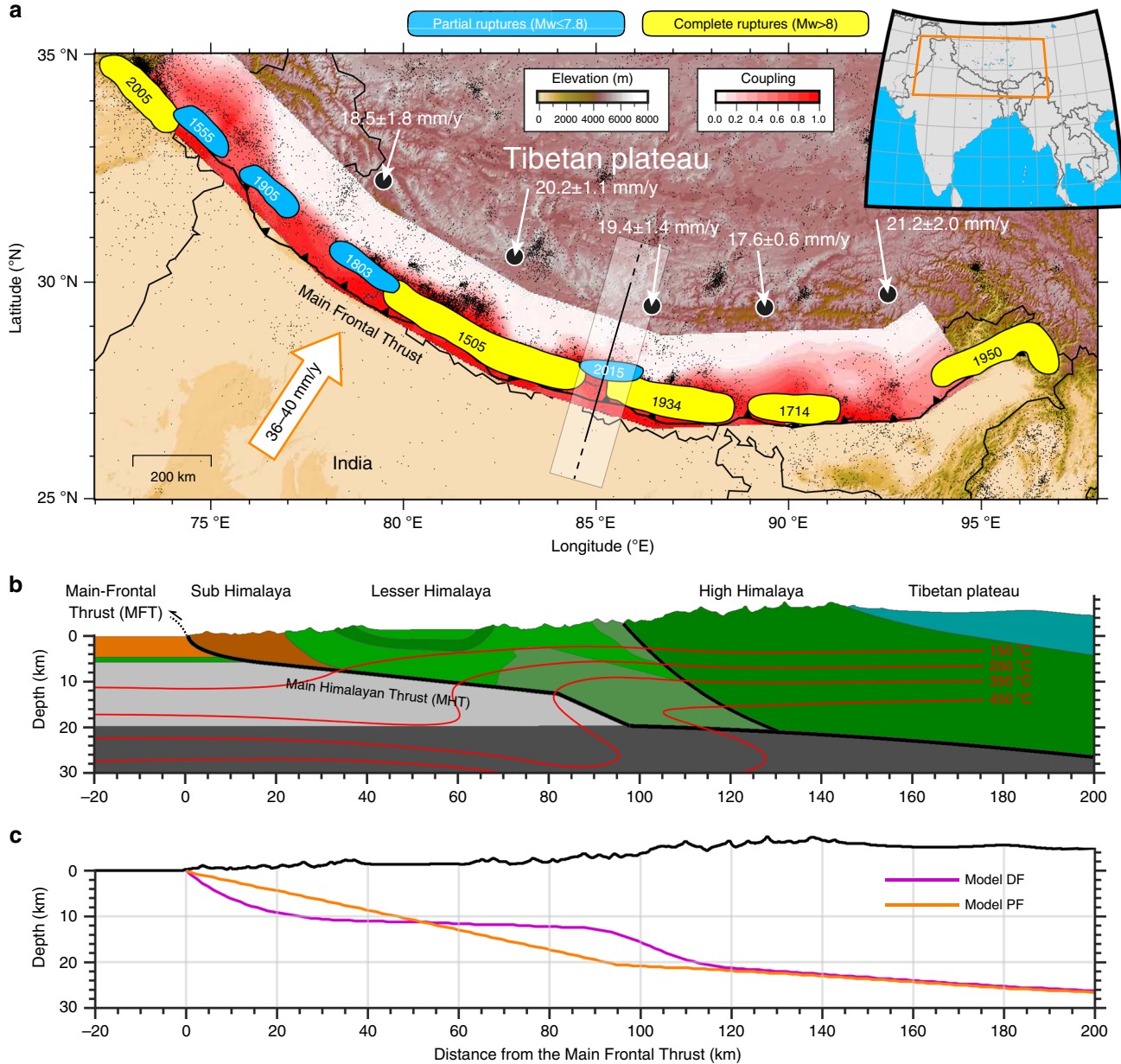

**Fig. 1** Seismotectonic context, model setup and fault geometries. **a**, Topographic relief, coupling mode and historical seismicity. The white arrows show the long-term shortening across the arc. The interseismic coupling is shown as shades of red (ref. [48]). A coupling value of 1 means the area is fully locked, while a value of 0 means fully creeping. Coloured patches indicate the supposed rupture zones since 1505 (refs. [4,21,22]): blue patches display blind ruptures of large (Mw ≤ 7.8) earthquakes, whereas yellow patches indicate surface ruptures of great (Mw > 8) events. Black line indicates the cross-section utilised for the numerical model setup. **b**, Zoom of the initial reference setup (model EF) and temperature. The numerical setup represents the geological cross-section of the Nepal Himalaya constrained from the main-shock and aftershocks of the Gorkha sequence (ref. [9]). **c**, Additional fault geometries employed in the numerical experiments: model DF, from Duputel et al.[10], and a planar fault geometry (model PF)

**Bimodal earthquake behaviour of the reference model**. Despite the 2D limitations, the reference model produces a rich earthquake behaviour, similar to that of natural faults. The spatiotemporal evolution of slip velocity of the reference model shows how coseismic slip events are released on the MHT fault (Fig. 3a). Although the whole seismogenic zone is interseismically nearly fully locked, most of the simulated earthquakes nucleate and propagate only in the lower edge of the locked Main Himalayan Thrust, whereas only a few events unzip the whole flat-and-ramp system. The largest events tend to have similar size and recur quasi-periodically every ~1250 years. Between them, a range of smaller events occurs, which release only small fraction of the

accumulated strain energy. Using a rupture width–moment magnitude scaling law[37], the moment magnitude of partial ruptures is estimated to Mw ~ 7.4–7.8 (Fig. 3b). Such cluster of differently sized partial ruptures leads up to a final complete failure of the MHT. These complete ruptures are the largest events with an estimated moment magnitude in the order of Mw ~ 8.3–8.4 (Fig. 3b).

To investigate the physical mechanism behind this behaviour, we analyse the spatiotemporal evolution of the stress and yield strength on the MHT. For example, event E9 (Fig. 3a) ruptures only the lower edge of the seismogenic zone and then event E18 is capable of propagating up to the surface. Our analysis indicates

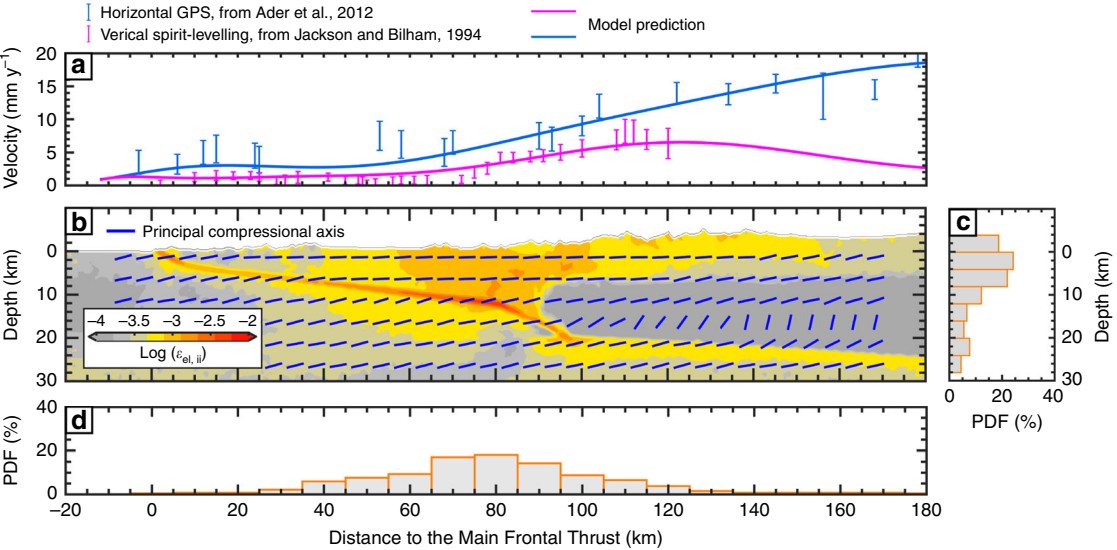

**Fig. 2** Interseismic behaviour computed in the 2D model. **a**, Observed vs. synthetic present-day velocity fields. Observed field (error bars) is shown in blue (horizontal GPS[8]) and violet (spirit-levelling[35]) bars, respectively. Solid lines show the corresponding horizontal and vertical modelling prediction. **b**, Elastic strain regime across the Himalaya inferred over an interseismic period of 350 years and orientation of principal compressional axes (blue bars). Histograms in **c** and **d** show the vertical and horizontal off-megathrust faulting distributions, respectively

that the partial rupture event E9 nucleates close to the downdip limit of the seismogenic zone, before the mid-crustal ramp, where the stress build-up due to tectonic loading is fastest (Fig. 3c). The rupture propagation causes a local stress drop (Fig. 3d), unzipping only part of the seismogenic zone as it is stopped as a result of a large initial strength excess—that is, difference between stress and yield strength. For this event, we further estimate the slip resulting from the occurrence of such rupture. Our results indicate that event E9 produces ~5–6 m of coseismic slip (Fig. 3e), mainly on the deeper flat portion of the MHT, between 10 and 15 km depth.

When only the downdip edge of the locked zone is unzipped, stress is transferred to the neighbouring updip region by a static stress transfer. Then the next downdip event nucleates sooner than expected from the average recurrence periods, with this new rupture being generally larger than the previous one. This occurs because the strength excess decreases in the frontal part of the MHT, as a result of the stress transfer and the ongoing tectonic loading. Consequently, partial ruptures contribute significantly to build up stresses to a critical level on the updip limit of the MHT, as for example before event E18 (Fig. 3f). Once strength excess is low throughout the MHT, a complete event eventually propagates through the whole ramp-flat-ramp fault system and leads to a large stress drop (Fig. 3g). These complete ruptures results in slip larger than 8 m (Fig. 3h), which is consistent with estimates from paleoseismic investigations[19,20,38]. Then a new cycle of partial ruptures begins, with an initial period of quiescence or small events activity (Fig. 3). This is exactly what our model shows in Fig. 3: temporal evolution of the MHT displays a bimodal seismicity-dominated regime. Notably, rupture events are triggered by stress build-up near the downdip end of the locked fault zone, as is observed in nature[39]. Also, the model reproduces a realistic earthquake sequence of irregular moment magnitude main shocks, including events similar to the 2015 Gorkha earthquake. A simulation example is shown in Supplementary Movie 1.

This bimodal pattern of large strength excess, low stress drop partial ruptures leading to infrequent low strength excess, high stress drop complete ruptures is also observed when analysing

their non-dimensional stress (S) parameter—that is, the ratio between the average strength excess before an event and the average coseismic static stress drop[40,41] (Fig. 4a–c). In terms of this S parameter, complete ruptures thus have relatively low values, while partial ruptures have relatively high values. When studying the kinematic slip evolution of each event, we observe both pulse- and crack-like ruptures (Supplementary Fig. 5). In a pulse-like mode, the local slip duration—also known as rise time—is much shorter than the total duration of the event[42]. On the contrary, shear cracks have an extended slip duration, even after the rupture has reached the surface, in which the rise time scales with final rupture width. Our models thus show that slip-pulses and shear cracks coexist along the same interface. They occur at marked points in time in relation to the stress state of the interface as defined by the S parameter. Large strength excess, prior to the event, leads to partial ruptures with relatively high S parameter that are pulse-like. These self-healing pulses could be the result of a strongly slip rate-dependent friction formulation that rapidly heals the fault for low characteristic slip velocities[43]. Conversely, relatively low S parameters are observed for complete ruptures that are crack-like. The hypothesis of a different rupture style for each event type is also supported by the identified mechanism of recurrent updip stress transfer toward a critical level[41,44], combined with results from dynamic rupture simulations that relate the fault stress state (S parameter) and rupture styles[45]. However, due to a low temporal resolution ($\Delta_t = 1$ year) and missing wave-mediated stress transfer, we cannot be sure that the actual stress states at the MHT are such that both crack- and pulse-like ruptures are feasible. In spite of these potential pitfalls, our models show similar features to those observed during the Gorkha earthquake. In particular, the 2015 main-shock was a pulse-like rupture with slip on any given portion of the fault occurring over a short fraction (~10 s) of the total ~70 s duration of the earthquake source[2].

A particular feature of the Himalayan wedge is the seismic–aseismic transition zone, which seems to coincide with the mid-crustal ramp beneath the front of the high Himalaya[8,11]. However, the feedback between the geometry and the rheological behaviour of the mid-crustal ramp are difficult to ascertain on the

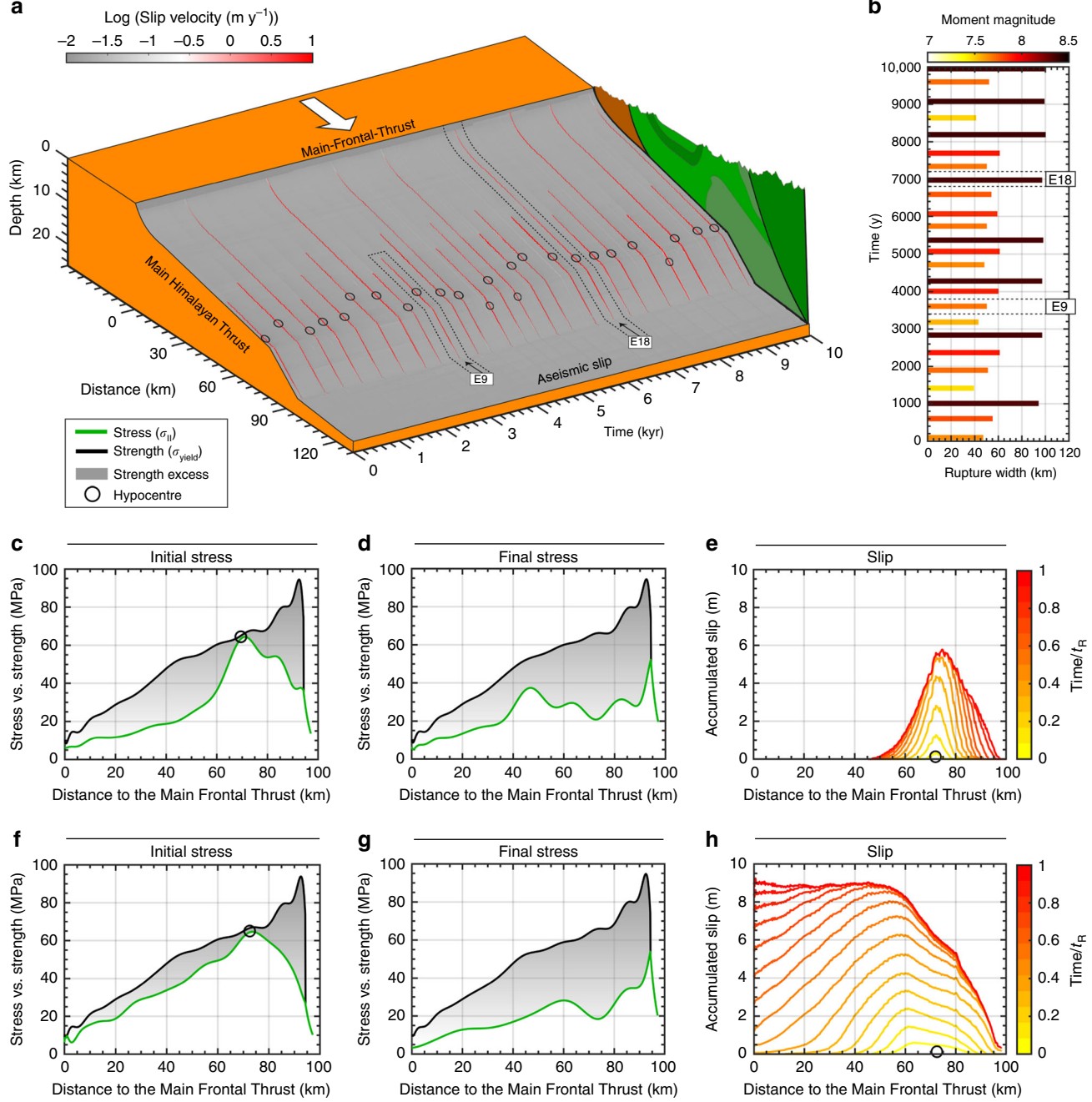

**Fig. 3** Megathrust behaviour computed in the 2D model (EF) over 10,000 years. **a**, Spatiotemporal evolution of slip on the MHT for the reference model. Red lines show slip during the simulated earthquakes. Note that hypocenters (black circles) are typically located in the lower edge of the flat segment, just before the mid-crustal ramp. **b**, Time evolution of downdip rupture width. Colorbar indicates the corresponding moment magnitude. **c**, **d**, Along megathrust profiles of initial (**c**) and final (**d**) stress vs. strength for the partial rupture event E9. **e**, Contours of accumulated coseismic slip throughout event E9. **f**, **g**, Along megathrust profiles initial (**f**) and final (**g**) stress vs. strength for the complete rupture event E18. **h**, Contours of accumulated coseismic slip throughout event E18

basis of natural observations alone. When a rupture occurs in our simulations, it generally expands upwards from the locked edge, but not much downwards. This occurs because the zone of aseismic slip acts as an efficient barrier to downdip propagation of ruptures. This self-consistent feature of our models as an effect of the temperature increase with depth, which in turn decreases the viscosity of rocks. Also, our models show that all hypocentre locations fall in a narrow zone near the edge of the mid-crustal ramp (Fig. 3a), indicating a pivotal role of this crustal asperity in localising the strain both on and off the megathrust (Fig. 2b).

Thus, our results suggest that both the geometric-structural and the thermal-rheological strength of the mid-crustal ramp control the downdip rupture width on the MHT.

**Effect of fault friction and geometry on seismic ruptures.** In our simulations we identify frictional properties and geometry of the MHT as key parameters that influence the emergence of the observed bimodal earthquake pattern. To examine the role of the three MHT fault geometries considered in this study, we first

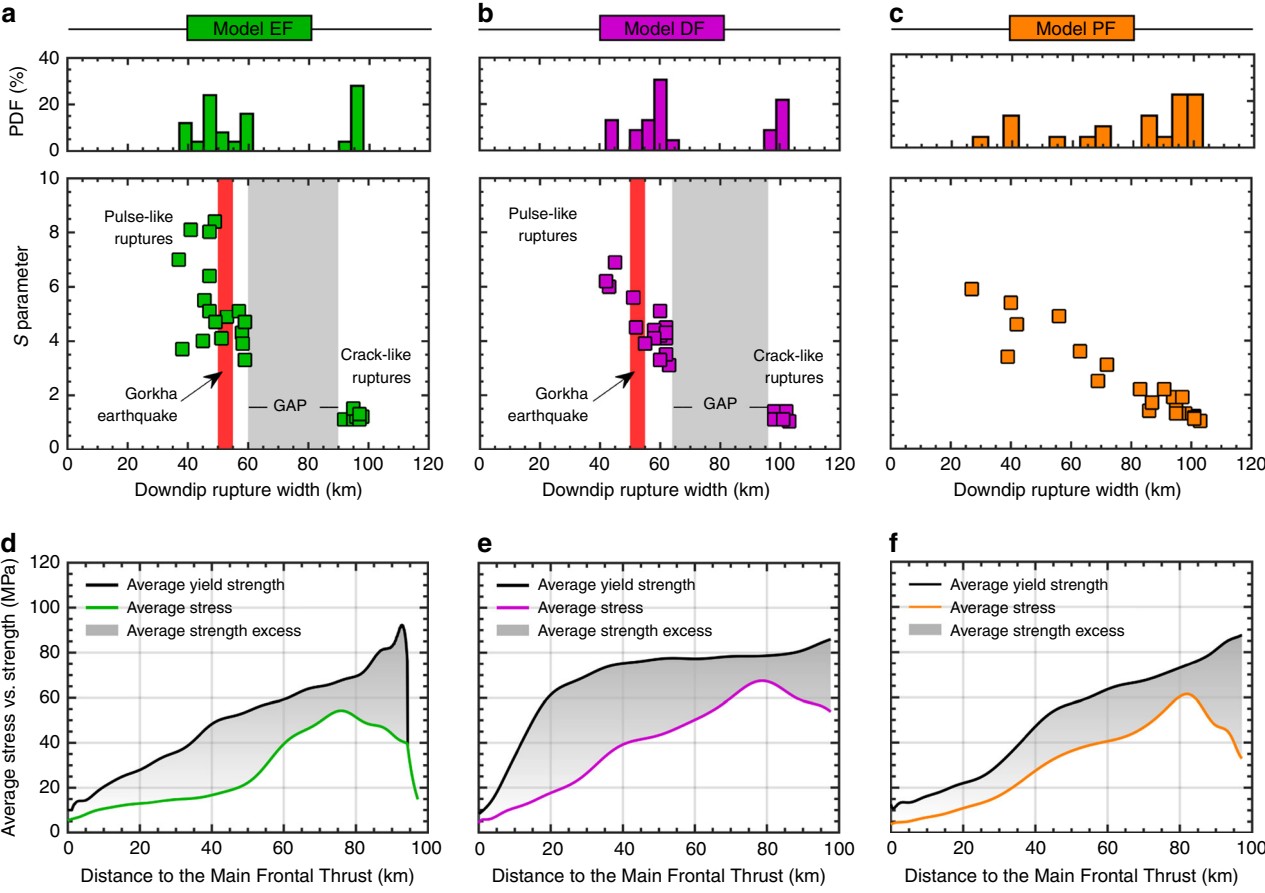

**Fig. 4** Impact of the three fault geometries on the rupture patterns. Relationship between the S parameter and rupture width for models adopting a realistic ramp-flat-ramp fault geometry inferred from Elliott et al.[9] (**a**) and Duputel et al.[10] (**b**), which also indicates the dominance of different rupture styles (pulse- vs. crack-like ruptures), and a planar fault geometry (**c**). **d–f**, Along megathrust profiles of the average stress vs. strength for the three fault geometries adopted: models EF (**d**), DF (**e**) and PF (**f**). Our simulations also indicate that, for each partial rupture, the S parameter is generally higher than in dynamic rupture simulations[45]. This results from the large amount of slip velocity-induced weakening ($\gamma$; see Methods section), which is motivated by laboratory experiments at coseismic slip rates[71]

analyse the relation between the S parameter and rupture width of all events when a bimodal seismicity pattern is observed ($\mu_s = 0.16$ and $\gamma = 0.7$; Fig. 4a–c). Results from the reference model EF (Fig. 4a) indicate that the S parameter decreases with increasing rupture width. Most importantly, we find that this ramp-flat-ramp geometry results in a rupture-width gap between 60–65 km and 90–95 km. A very similar trend is also observed in model DF (Fig. 4b). Pulse-like partial ruptures are confined to a critical width of 60–65 km, whereas large crack-like events propagate through the whole seismogenic zone. Consequently, models EF and DF result in a bimodal distribution of rupture widths. On the other hand, results from the simple planar fault (Fig. 4c) indicate that the S parameter decreases linearly with increasing rupture width. This means that the larger the event, the higher the stress released and the resulting S parameter is lower. Although this model displays a wide spectrum of rupture widths, the general pattern does not indicate any bimodal distribution.

We then analyse the average downdip stress vs. strength distribution for the three fault geometries adopted (Fig. 4d–f). In general, these profiles suggest that, the steeper the fault dips in the updip region of the MHT, the higher would be the pressure-dependent fault strength. This, together with a relatively higher fault friction, increases the fault strength even further. Consequently, the strength excess also increases, and a higher pre-stress is thereby necessary to reach a critical level at which eventually a

crack-like event ruptures the entire megathrust. As in the case of model EF (Fig. 4d), and even more clearly on model DF (Fig. 4e), the strength excess in the shallower region of the MHT is notably high. This behaviour arises because when the model accounts for a ramp-and-flat fault geometry, the far-field tectonic loading is not fast enough to bring the pre-stress up to a critical state in the upper edge of the MHT. Most of the simulated earthquakes are thus capable of rupturing only a fraction of the seismogenic zone. Then, the static stress distribution left over from these previous partial ruptures contribute significantly to increase the stress state in the updip limit of the MHT. On the other hand, the planar fault geometry (model PF) maintains a relatively low strength excess throughout the seismogenic zone (Fig. 4f), thereby allowing the propagation of frequent complete ruptures.

We then explore the effect of static fault friction ($\mu_s$), and of the maximum friction drop from static to dynamic friction coefficient ($\gamma = 1 - \mu_d/\mu_s$) on the resulting bimodal pattern of large earthquakes. Our model produces distinctly different rupture patterns within a narrow range of frictional parameters (Fig. 5). In fact, an increase of both the static fault friction and friction drop leads to an increase of the number of events per cycle (Fig. 5a), the average recurrence interval between the largest events (Fig. 5b), and the median S parameter values (Fig. 5c). As illustrated in Fig. 5, this corresponds to a transition from ordinary (unimodal) cycles to irregular cycles, which display a bimodal seismicity (see

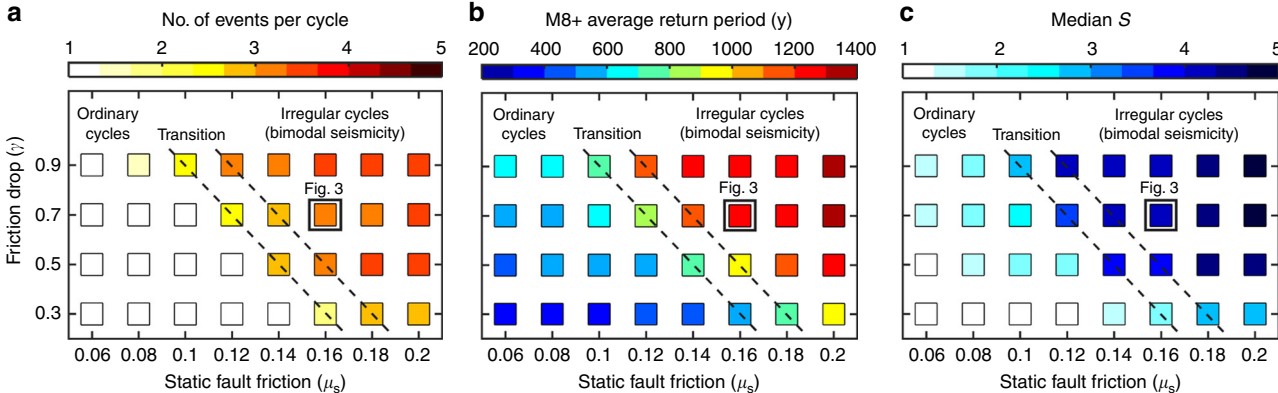

**Fig. 5** Effect of frictional properties on the seismic behaviour of model EF. Average **a** number of events per cycle, **b** recurrence time of complete ruptures (Mw > 8 events) and **c** median of the S parameter. Dashed black lines indicate the transition from ordinary cycles to irregular cycles (bimodal seismicity)

also Supplementary Fig. 6). In contrast, the spatiotemporal evolution of the model with a lower static fault friction ($\mu_s = 0.1$) shows a more ordinary recurrence pattern of quasi-periodic large events (Supplementary Fig. 7). These events mostly nucleate near the edge of the mid-crustal ramp, propagate both up- and downwards, and typically activate the whole flat-and-ramp system. These ruptures break the entire locked zone of the MHT in a crack-like style, and lead to significant stress drops. Consequently, this model is related to a low median S value (Fig. 5c). Thus, these results indicate that the maximum friction drop ($\gamma$) can significantly affect the recurrence interval of complete ruptures (Fig. 5b) and the average S parameter (Fig. 5c), but it cannot prevent the genesis of bimodal seismicity for relatively higher fault friction ($\mu_s \geqslant 0.16$).

## Discussion

Our simulations show that it is probably incorrect to assume that earthquakes known to have occurred along the Himalayan front over history[46] are representative of the greatest possible earthquakes. Along many segments, the large historical events probably represent partial ruptures with magnitude significantly lower than what complete ruptures would produce. In our model, the same segment of the MHT can in principle produce a sequence of partial ruptures similar to the Gorkha earthquake and occasionally much larger events, similar to the 1934 Mw 8.4 event or even larger. This is confirmed by moment conservation calculations at the scale of the Himalayan arc, which require Mw ~ 9 earthquakes with a 800–1000 years return period[47]. Our models indicate that a great earthquake (Mw > 8) can occur at the same location as a Mw ≤ 7.8 earthquake, and that it may strike sooner than would be anticipated from considerations of renewal time from plate convergence rates. While we cannot rule out the plausible presence of along-strike heterogeneities given the lack of the third dimension, our models show that the combined effects of fault geometry and frictional properties in controlling the along-dip bimodal behaviour of the MHT could potentially hold for the entire Himalayan arc. In support of this claim, recent pattern of interseismic coupling on the MHT along the entire Himalayan arc do not indicate any aseismic barrier that could affect the seismic segmentation of the arc and limit the along-strike propagation of seismic ruptures[48] (Fig. 1a).

For a finite range of static fault friction ($\mu_s = 0.06$–0.2), our model exhibits a large spectrum (250–1500 years) of recurrence time of great earthquakes. It also shows that an indication for the temporal proximity of such a Mw > 8 earthquake can come from the maximum updip limit of the prior, partial earthquake, which

provides an indication for a likely critically stressed MHT (Fig. 3a). Our results indicate that an average recurrence time of ~600 years leads to coseismic slip of 8–10 m in order to release the elastic strain accumulated during such interseismic periods. However, partial ruptures account only for an average slip of 4–6 m, in agreement with the average slip of moderate (Mw ≤ 7.8) Himalayan earthquakes such as the Gorkha earthquake[9].

Finally, it appears that the static stress change due to partial ruptures is the major factor introducing irregularity in the seismic cycle. This is the main reason that could explain why the model obeys neither the slip- nor time-predictable behaviour at any given point on the fault (Supplementary Fig. 8), since it does not incorporate a fixed threshold shear stress for slip to occur. This is because, after each earthquake, the stress on the ruptured area drops to a low level, approximately determined by the rate-dependent friction formulation evaluated at the coseismic slip rate.

To conclude, this seismo-thermo-mechanical model constrained by observations provides physical explanations to understand the behaviour of the seismic cycle in the Himalaya. It shows that frictional properties and non-planar geometry of the MHT control a variety of phenomena, such as the bimodal seismicity, the relative persistence of along-dip variations of seismic ruptures, and the variable recurrence time of large (Mw ≤ 7.8) and great (Mw > 8) earthquakes. Based on our numerical experiments, we postulate that large crack-like earthquakes on the MHT may release stress inherited from former pulse-like partial ruptures. These very large events account for the bulk of the deformation that is transferred to the most frontal structures in the Sub-Himalaya. If this scenario is in fact correct, it has consequences for the assessment of seismic moment where only rupture length and surface slip are known, as is the case for all palaeoseismic ruptures inferred from slip on the MFT[16,19,20,22]. Because a heterogeneous strain condition is likely to prevail throughout the Himalaya, our results may provide an answer to the long-standing difficulties in explaining the source of the stored stresses needed to drive large (>8–10 m) paleoseismic surface ruptures recorded on the MFT[20,49]. The risk of such extreme earthquakes may have been underestimated because of the evidence that these events might only exist in geological records given their millenary return period[47]. Seismicity catalogues might also give the false impression that they include the largest possible earthquakes. This might be related to the magnitude gap separating Mw 8+ events from the dominant mode formed by the smaller partial ruptures, which make up the bulk of the seismicity.

In light of our modelling results, the updip arrest of the 2015 Gorkha earthquake calls for special attention. That fault patch, updip of the Gorkha rupture, stayed locked in the post-seismic period[50]. The stress level was increased by the Gorkha earthquake[2], making that patch more likely to fail in the next large rupture of the MHT in that area. The nearly 800-km-long stretch between the 1833/2015 ruptures and the 1905 Mw 7.8 Kangra earthquake is also a well-identified seismic gap with no large earthquake for over 500 years[1]. The MHT is clearly locked there[8,48] and its deficit of slip may exceed ~10 m. The last large earthquake in that area occurred in 1505, and could have exceeded Mw 8.5[51]. These factors make this area a prime location for a future complete rupture of the MHT. Continued geodetic monitoring of the Himalayan arc in the coming years will help to provide new constraints and to ascertain these speculations.

## Methods

**Seismo-thermo-mechanical methodology**. The 2D seismo-thermo-mechanical (STM) code uses an implicit, conservative finite difference scheme on a fully staggered Eulerian grid in combination with a Lagrangian marker-in-cell technique[52]. The code solves for the conservation of mass for an incompressible material, momentum and energy. The advection of transport properties including viscosity, plastic strain and temperature is performed with the displacement of Lagrangian markers. The following three mechanical equations are solved to obtain the horizontal and vertical velocities, $v_x$ and $v_z$, and pressure $P$ (defined as the mean stress):

$$\frac{\partial v_x}{\partial x} + \frac{\partial v_z}{\partial z} = 0 \tag{1}$$

$$\frac{\partial \sigma'_{xx}}{\partial x} + \frac{\partial \sigma'_{xz}}{\partial z} - \frac{\partial P}{\partial x} = \rho \frac{Dv_x}{Dt} \tag{2}$$

$$\frac{\partial \sigma'_{zz}}{\partial z} + \frac{\partial \sigma'_{zx}}{\partial x} - \frac{\partial P}{\partial z} = \rho \frac{Dv_z}{Dt} - \rho g \tag{3}$$

where $\rho$ is density, $\sigma'_{ij}$ are deviatoric stress tensor components, and $g = 9.81 \text{ m s}^{-2}$ is the vertical component of the gravitation acceleration. The momentum equations include the inertial term to stabilise high-coseismic slip rates at low time steps. A time step of 1 year, however, reduces our formulation to a virtually quasi-static one. Ruptures during the resulting events hence represent the occurrence of rapid threshold-exceeding slip during which permanent displacement and stress drop occur along a localised interface. The energy equation describes the balance of heat in a continuous medium and relates temperature changes due to internal heat generation, as well as with advective and conductive heat transport[53]. The Lagrangian form of energy equation solves for the temperature $T$:

$$\rho C_p \frac{DT}{Dt} = \frac{\partial}{\partial x_i}\left(k\frac{\partial T}{\partial x_i}\right) + H_r + H_s \tag{4}$$

where $C_p$ is isobaric heat capacity, $k$ is thermal conductivity, $H_r$ is radioactive heat production and $H_s$ is shear heat production during non-elastic deformation, as follows:

$$H_r = \text{cst.}; \tag{5}$$

$$H_s = \sigma'_{ij}\,\dot{\varepsilon}_{ij,vp}; \tag{6}$$

where $H_r$ is a constant value for each rock type, $\dot{\varepsilon}_{ij,vp}$ is the visco-plastic component of the deviatoric strain rate tensor.

**Rheological model**. The fundamental Eqs. (1)–(4) are solved using constitutive relations that relate deviatoric stresses and strain rates in a nonlinear visco-elasto-plastic manner:

$$\dot{\varepsilon}_{ij} = \frac{1}{2G}\frac{D\sigma'_{ij}}{Dt} + \frac{1}{2\eta}\sigma'_{ij} + \begin{cases} 0 & \text{for } \sigma'_{II} < \sigma_{yield} \\ \chi\frac{\partial \sigma'_{ij}}{\partial \sigma'_{ij}} = \chi\frac{\sigma'_{ij}}{2\sigma'_{II}} & \text{for } \sigma'_{II} = \sigma_{yield} \end{cases} \tag{7}$$

where $G$ is shear modulus and $\eta$ is effective viscosity. $D\sigma'_{ij}/Dt$ is the objective co-rotational time derivative solved using a time explicit scheme[53] and $\sigma_{II} = \sqrt{\sigma'^2_{xx} + \sigma'^2_{xz}}$ is the second invariant of the deviatoric stress tensor, and $\chi$ is a plastic multiplier connecting plastic strain rates and stresses. Introducing a visco-plastic viscosity ($\eta_{vp}$), we can rewrite Eq. (7) as:

$$\sigma'_{ij} = 2\eta_{vp}Z\dot{\varepsilon}_{ij} + \sigma_{ij}(1 - Z) \tag{8}$$

where $Z$ is the visco-elasticity factor:

$$Z = \frac{G\Delta t_{comp}}{G\Delta t_{comp} + \eta_{vp}} \tag{9}$$

where $\Delta t_{comp}$ is the computational time step.

The plastic behaviour is taken into account assuming a non-associative Drucker–Prager yield criteria[54]. Plastic flow is evaluated at each Lagrangian marker if $\sigma'_{II}$ reaches the local pressure-dependent yield strength $\sigma_{yield}$:

$$\sigma_{yield} = C + \mu_{eff}\,P \tag{10}$$

where $C$ is the cohesion.

An important component in the yield criterion is the friction coefficient. Following the approach of van Dinther et al.[55], we apply a strongly rate-dependent friction formulation[56], in which the effective friction coefficient $\mu_{eff}$ depends on the visco-plastic slip velocity $V = (\sigma_{yield}/\eta_m)\Delta x$, in which $\eta_m$ is the local viscosity from the previous time step and $\Delta x$ is the Eulerian grid size:

$$\mu_{eff} = \mu_s(1 - \gamma) + \mu_s\frac{\gamma}{1 + \frac{V}{V_c}} \tag{11}$$

$$\gamma = 1 - (\mu_d/\mu_s) \tag{12}$$

where $\mu_s$ and $\mu_d$ are static and dynamic friction coefficients, respectively, $V_c$ is the characteristic velocity, namely the velocity at which half of the friction change has occurred, and $\gamma$ represents the amount of slip velocity-induced weakening if $\gamma = 1 - (\mu_d/\mu_s)$ is positive, or strengthening if $\gamma$ is negative.

When plastic yielding condition is locally reached, we require a constant second invariant of deviatoric stresses (assuming the absence of elastic deformation).

$$\text{If } \sigma'_{II} = \sigma_{yield} : \left\{ \frac{D\sigma'_{II}}{Dt} = 0, \quad \dot{\varepsilon}_{ij}^{\ elastic} = 0 \right\} \tag{13}$$

then the stress components are similarly (i.e., isotropically) corrected so that

$$\sigma'_{ij} = \sigma'_{ij}\cdot\frac{\sigma_{yield}}{\sigma'_{II}}. \tag{14}$$

Accordingly, the local viscosity-like parameter $\eta_{vp}$ decreases to weaken the material and to localise deformation

$$\eta_{vp} = \eta\,\frac{\sigma'_{II}}{\eta\chi + \sigma'_{II}} \tag{15}$$

where

$$\chi = 2(\dot{\varepsilon}_{II} - \dot{\varepsilon}_{II}^{\ viscous}) = 2\left(\dot{\varepsilon}_{II} - \frac{1}{2\eta}\sigma'_{II}\right) \tag{16}$$

and

$$\dot{\varepsilon}_{II} = \sqrt{\dot{\varepsilon}_{xx}^{\ 2} + \dot{\varepsilon}_{xz}^{\ 2}}. \tag{17}$$

Finally, the visco-plastic viscosity $\eta_{vp}$ is corrected during plastic deformation:

$$\eta_{vp} = \frac{\sigma_{yield}}{2\dot{\varepsilon}_{II}}. \tag{18}$$

On the other hand, if the plastic yielding condition is not satisfied, this means that the material is under elastic and/or viscous deformation (i.e., diffusion and/or dislocation creep), therefore $\eta_{vp} = \eta$.

**Model setup and boundary conditions**. The initial 2D model setup consists of a $1000 \times 250$ km computational domain (Supplementary Fig. 1). The visco-elasto-plastic thermomechanical parameters of these lithologies are based on a range of laboratory experiments and are listed in Supplementary Table 1. The models use a grid resolution of $1491 \times 404$ nodes with variable grid spacing. This allows a high resolution of 200 m in the area subjected to largest deformation. More than 35 million Lagrangian markers carrying material properties were used in each experiment. Velocity boundary conditions are free slip with the exception of the permeable lower boundary along which infinity-like external free slip and external constant temperature conditions are imposed implying free slip and constant temperature condition to be satisfied at 1000 km below the bottom of the model[57]. The free surface boundary condition atop the crust is implemented by using a 'sticky-air' layer[58] with low density (1 kg m$^{-3}$) and viscosity ($10^{17}$ Pa s). A pre-scribed convergence rate of 38 mm year$^{-1}$ is imposed on the left boundary, as inferred from several GPS campaigns[59]. This allows the subducting Indian plate to converge underneath the Asian upper plate. The model accounts for shear heating (see Methods, Eq. (6)) and for solid–solid phase transitions, such as the process of eclogitization, which has been shown to be an important component of the overall buoyancy budget of the underthrusting Indian lower crust[60]. These phase

transitions are parameterised as a function of thermodynamic state variables (P, T, V) and composition by using polynomials to interpolate the reaction boundary[61].

**Initial geometry**. Our proposed initial geometry of the India–Asia collision is based on the crustal data[62] and geophysical constraints[7,63,64], and is also consistent with geomorphic and geologic structural constraints[6,8,9,31,60,65]. The initial setup consists of a ~600-km-long Indian plate (on the left, Supplementary Fig. 1) underthrusting the fixed overriding Asian plate (on the right) and the Himalayan wedge. The Indian upper crust is made of a thick layer of sediments (4-km thick) and crystalline rocks (16-km thick), overlying a 16-km-thick lower crust and a 65-km-thick lithospheric mantle. On the other side, the Eurasian plate is made of a very thick upper crust (36-km thick) and lower crust (30-km thick) due to crustal shortening and thickening. The tectonic architecture of the Himalayan wedge is instead more complicated, since it represents the impingement of the Indian continental margin on the Eurasian plate. The Himalayan fold-and-thrust belt is here divided into four lithotectonic units: the Sub-Himalaya, the Lesser Himalayan, the Higher Himalaya and the Tibetan Plateau. The geometry of the MHT across the Himalaya has been derived from geophysical observations and surface geology. The frontal part of the fault system is well constrained from balanced cross sections across the Siwalik fold-and-thrust belt and some gravity and magnetotelluric data[6,66]. These data show that the MFT, the MBT and the intervening thrust faults all root into a 5–7 km deep décollement at the top of the underthrusted Indian basement. The décollement probably extends beneath the front of the Higher Himalaya, as indicated from a zone of high conductivity that has been interpreted as sediments dragged along the décollement[65]. This décollement constitutes the sole of the Himalayan wedge and is usually called MHT fault. The décollement extends beneath the Lesser Himalaya at depth of 7–12 km, and connects with a mid-crustal ramp (12–22 km depth) beneath southern Tibet. This mid-crustal ramp has been proposed in a number of previous studies[30,39,67] and is in agreement with recent slip inversions after the Gorkha event[9]. This deeper ramp then roots into a shallow north-dipping shear zone of aseismic deformation, coinciding with the deeper portion of the MHT imaged from various seismological experiments[7,63]. This complicated 2D geometry is then transferred into the computational domain by using GeoMAC (Geo-Mapping And Converting tool), a C–based tool that allows for resampling and converting drawings into material properties. Hence, GeoMAC converts 2D vector drawings based on geological profiles, seismic data, tomography models and other geophysical constraints, by assigning material phases to Lagrangian markers inside the large-scale tectonic shapes.

**Friction properties on the Main Himalayan Thrust**. Several constraints indicate that the friction on the MHT is low[31]. For instance, recent thermometric and thermochronological data from the central Nepal suggest that the shortening across the range has been taken up primarily by thrusting along the Main Himalayan Thrust fault, with negligible internal shortening of the Himalayan wedge[13]. These data thus suggest that friction along the MHT is ~0.07, which is in agreement with the observed pattern of erosion and the present morphology of the Himalayan range[12,31]. Also, given the slope of the Himalayan wedge and the dip angle of the MHT beneath the Lesser Himalaya, the effective basal friction on the flat portion of the MHT is inferred to be smaller than 0.12[30]. This is in keeping with previous numerical models, which require a basal effective friction <0.13. This value is consistent with the previous analysis of Davis et al.[68] that, considering the whole Himalayan wedge (which has a steeper slope), results in a larger effective basal friction of 0.25. Another constrain comes from relationship between the topographic elevation and the cutoff in the microseismic activity, which is used to infer the shear stress on the fault. Since the décollement lies at a depth of ~10 km and the inferred shear stresses cannot exceed 35 MPa, this corresponds to a friction of ~0.1[39].

**Temperature distribution**. The thermal structure of both Indian shield and Asia (Supplementary Fig. 1c) are calculated from the steady-state continental geotherm[69]. The initial thermal gradient is set using values of 273 K at the top and 1617 K at the bottom of the lithosphere, whereas within the mantle was quasi-adiabatic (0.5 K/km). The thermal structure within central Nepal is computed using the thermokinematic model proposed in Bollinger et al.[38] and updated in Herman et al.[13]. The formal inversion suggests a radiogenic production of 2.2 $\mu Wm^{-3}$ for the Higher and Lesser Himalaya units and a value of 0.8 $\mu Wm^{-3}$ for the middle and lower crust of India[13], which are used for the corresponding lithologies of our numerical model. This latter was obtained from a formal inversion of the large data set of thermochronological and thermobarometric data available from central Nepal. The topography is assumed to be steady state. The model fits the data and shows a transition from brittle to ductile rheologies at a temperature of about 400–450 °C, corresponding to about 18-km depth on the Main Himalayan Thrust.

**Modelling procedure**. The STM modelling approach adopted for this study comprises two steps. Prior to the first modelling stage, we define the initial geometry, rock properties, temperature distributions and the boundary conditions. During the first stage, which utilises a time step of 100 years, the stress builds up and all the physical properties iterate up to reach a isostatic equilibrium. Stress build-up occurs as the strain is accumulated. Differential loading due to rheological discontinuities, tectonic asperities, temperature and viscosity distribution causes heterogeneous stress localisation in the Himalayan wedge (Fig. 2). In the second modelling stage, the time step progressively decreases to approach a final value of 1 year, while the inertia term and rate-dependent friction are activated. During the seismic cycle, when the maximum strength is locally reached, the instability is fed through the feedback of decreasing viscosities. This increases slip velocities, which decreases the slip rate-dependent friction and strength, and in turn decreases viscosities even further. Spontaneous rupture propagation occurs because stresses are increased ahead of the rupture front to balance the dropping stresses within the rupture and to thereby maintain a static equilibrium. Finally, healing of strength occurs as slip velocities decrease.

**Modelling limitations**. General limitations of our modelling approach are discussed in previous studies, in which the STM models have been applied to subduction[55] and collision zones[70]. In nature, earthquake ruptures occur within a three-dimensional, geometrically complex fault system with various scales of downdip and along-strike variations in its seismogenic behaviour. The lateral, third dimension is absent in our numerical model. That means that our two-dimensional plane strain model ignores lateral variations in interseismic stress build-up and rupture propagation. Compared to nature, our model produces unrealistically long seismic events because of the large time step (i.e., 1 year). This means that a simulated event or earthquake refers to the occurrence of rapid threshold-exceeding slip during which permanent displacement and stress drop occur along a localised interface. On the other hand, the presented results generally demonstrate a satisfactory agreement with a wide range of long- and short-term natural observations in the Himalaya. Events in our numerical model have reasonable downdip extension, similar behaviour and comparable surface displacement. However, it is important to stress that our incompressible inertial formulation does not account for the full inertial dynamics. Also, we acknowledge that modelling pressure and shear waves might impact part of our results. In spite of these limitations, our simulations mostly cover new ground, as yet unexplored, not only as far as the bimodal seismicity is concerned, but also how short-term seismic processes are related to the long-term interseismic deformation.

**Code availability**. Computer code used within the manuscript and its Supplementary Information are available from the corresponding author upon reasonable request.

## Data availability
Data within the manuscript and its Supplementary Information are available from the corresponding author upon reasonable request.

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

## Acknowledgements

This study was funded by the SNF 2-77090-14 project Swiss-AlpArray SINERGIA. We gratefully acknowledge A.A. Gabriel, E. Kissling, J.-P. Ampuero, L. Bollinger, R. Jolivet, G. Hetényi, R. Almeida, S. Barbot and the STM-group for comments. V.L. Stevens kindly

provided the coupling data. We are grateful to J. Singer for providing us with a basic GMT script to plot Fig. 1a. Numerical simulations were performed on ETH cluster Euler.

## Author contributions

L.D.Z. designed the study and model setup, carried out and analysed the numerical experiments, and wrote the paper. Y.v.D. and T.G. developed the STM methodology and analysed the results. J.-P.A. contributed to the concept development. All authors discussed the results and commented on the paper.

## Additional information

**Competing interests:** The authors declare no competing interests.

