## [Peer Review File · Nature Communications]

Reviewers' comments:

Reviewer #1 (Remarks to the Author):

Review of "Bimodal seismicity in the Himalaya controlled by fault friction and geometry," by Dal Zilio et al.

This is a very interesting manuscript that tackles a problem of extreme societal importance: the earthquake potential of the Himalaya, and in particular the vicinity of the recent Gorkha, Nepal, earthquake. The authors use a new and innovative modeling method to simulate multiple seismic cycles (albeit in 2D) in a system that includes a tremendous number of realistic details, such as fault geometry, thermo-mechanical properties, and plastic failure. They experiment with a range of fault geometries and frictional properties to investigate which combinations lead to the observed bimodal pattern of seismicity in this area. The results may have important implications for our estimates of earthquake hazard—in particular, the authors' models imply that there may be plenty of stress left over in the Gorkha region for a much larger earthquake than the recent one, threatening Kathmandu.

Overall I think this is a well-designed study that should be published. I have some big questions about the method, though these questions probably betray my lack of familiarity with this type of modeling. It still is not entirely clear to me what constitutes a "fault" in the current method. I gather from the method description that rather than having slip on a surface, the earthquakes actually correspond to shear failure of a zone of finite width that fails in some sort of volumetric shear. The authors are able to define effective coefficients of friction, but they use static coefficients that are much smaller (<0.2) than are commonly used in "standard" spontaneous dynamic rupture models. Their S values are also quite high compared to those used in previous faulting studies. It also was not clear to me the time scale of the rupture events the authors are modeling. At one point the authors note that they have a time step of 1 year and that they have effectively a quasi-static formulation, but yet they make a point of their method including inertia. How this relates to the rupture and slip propagation documented in Figure S4 is not clear, because the times are scaled by the overall time of the rupture.

None of these questions are meant to be criticisms of the method—I am pleased that the researchers are applying a novel one, because it indeed looks like a reasonable way to model the seismic cycle. However, to be useful to a broader audience (in particular, one that is used to different earthquake modeling methods), the authors should take more care to lay out exactly how their method works, particularly during the dynamic rupture process, and argue more strongly for their choice of frictional parameters.

I have a number of other, more minor comments, that I have included on annotated versions of manuscript and supplementary materials.

I recommend the paper be published with minor revisions.

Reviewer #2 (Remarks to the Author):

This study uses a sophisticated thermo-mechanical model with elasto-visco-plastic rheology and velocity-weakening frictional plasticity to examine earthquake behavior on the MHT under the Himalaya. In particular, the authors focus this paper on the "bimodal" behavior of earthquake ruptures in context of this model. Bimodal earthquake behavior is suspected to be the long-term behavior of the natural system based on historic and paleoseismic records of earthquakes.

This is indeed a very important problem in earthquake science and it is a fascinating setting with

wonderful observations. The work in this paper is very impressive and this study makes an important contribution to the “debate” about bimodal seismicity and the geometry of the MHT. The natural evolution of bimodal events in the numerical simulations is clearly an important result.

I really like this study, and I am very impressed by the sophistication of the simulations and the intent to model earthquakes in a system where the stress and (to some extent) the thermal and rheological conditions evolve naturally. The authors have gone to great lengths to generate a mechanical system that is consistent with a wide range of geophysical observables and the model setting in Figure 2 is quite impressive – the model is able to match first-order geodetic patterns as well as spatial distribution of off-MHT seismicity (although the seismicity is not shown, but it is described). It is however, very difficult to assess this model and digest all of Figure 2 because all of the methodology is buried in Supplementary materials and the Methods document. I understand this is the “nature” of Nature papers, but it does make it quite difficult to understand the technical aspects of this paper.

For this paper, the authors effectively treat Figure 2 and all the model development/conditions to generate Figure 2 as the backdrop, or setting for the main point of the paper, which is how earthquakes naturally evolve in this mechanical system. I have no problem with this, but in my opinion, Figure 2 and the supplementary figures required to understand how Figure 2 could (perhaps should?) be a stand-alone paper itself. It is very impressive work. I emphasize that it is impossible to understand this Figure without working through the Methods and Supplementary sections, which again, is perhaps not a problem, but it requires the reader to work very hard.

I do think it would be necessary to at least have a short paragraph in the main text describing the model setup which is currently entirely in the Supplementary information. In fact, Figures 2 and 3 are presented without any mention of friction parameters. It is necessary for the reader to know just a few basic conditions to understand Figures 2 and 3: friction parameters and spatial distribution (are the friction parameters the same on and off the MHT, for example?); basic boundary conditions, like how shortening is imposed; how is the MHT distinct from the bulk medium – why does deformation localize on the MHT? It’s not even clear whether these are dynamic simulations or quasi-static.

One issue I am left wondering about after reading this paper is what, ultimately, is the physics controlling the rupture timing and spatial extent in this model? This is the entire point of the paper. Do you really need the sophisticated thermo-mechanical model to produce the bi-modal earthquake behavior, or does it all come down to the choice of friction parameter and friction drop, as illustrated in Figure 4? Would you get this bimodal behavior in, say, an elastic halfspace without any off-fault inelasticity? In the end, it seems the bimodal behavior is controlled entirely by the friction parameters and geometry. And from Figure 5 it appears that the bend in the fault is ultimately what is needed to get the truly bimodal distribution. I wonder if this bimodal behavior would arise from any model with some slip weakening and the mid-crustal ramp geometry?

Some more specific questions/comments:

1. In the abstract lines 5-8. This sentence is awkward and confusing. You have a colon followed by a statement about the nature of bimodal seismicity. However the way this sentence reads to me, I would expect the phrase after the colon to be about the factors that regulate seismicity.
2. Abstract lines 12-16. I know you have the word “may” in there, but these statements sound like you are saying that next rupture is likely to be much larger than the Gorkha, but your simulations suggest several Gorkha-size earthquakes can occur in sequence. I don’t think you can say anything about the next earthquake – only that you can expect Gorkha-sized events more frequently than the M8+ events.
3. Line 36. You are using “mid-crustal” as a noun. Is that intentional? It looks like an adjective.
4. Line 62. I don’t think you ever explicitly stated that the EF geometry is shown in Figure 1b.
5. Lines 80-84. It is unclear what you mean by “shortening on a non-planar dislocation”? You can’t

have shortening on a dislocation. Why do you have 19-20 mm/yr of convergence across the Himalaya when you impose 38 mm/yr of shortening according to the Methods section? Are you saying you impose 38 mm/yr of total shortening and 19-20 mm/yr occurs as slip across the MHT. Where does the rest of the shortening go? This needs to be cleaned up, I think.

6. Figure 2 shows elastic, recoverable strain. How much strain is permanent, inelastic? It would be interesting to see that, although I realize it isn't the purpose of this paper.

7. In Figure 2, what component of strain is shown? I can't understand the colorbar label. Why is "interseismic strain" so localized on the MHT? If this fault is "locked" interseismically, I would expect the elastic strain to be quite diffuse. Is the fault zone more compliant (elastically) than the surrounding medium?

8. Line 90 is an awkward sentence. It doesn't make sense. What is "mechanical consistency"? Consistent with what?

9. Line 103. I mentioned this previously, but you have not given any information about the conditions in the "reference model" (friction, etc.).

10. Are friction parameters, μ_s and μ_d , uniform with depth on the fault?

11. Why do you consider such very low friction coefficients? Coefficients less than 0.2 are considered quite low for rock. I believe some clay minerals and phyllosilicates have coefficients of friction as low as 0.2, but typically friction is much higher. Are you considering this as an effective coefficient of friction that includes pore pressure?

Bimodal seismicity in the Himalaya controlled by fault friction and geometry

Luca Dal Zilio, Ylona van Dinther, Taras Gerya and Jean-Philippe Avouac

Response to the Reviewer's Comments

September 7, 2018

>> Reviewers' comments:

Reviewer #1 (Remarks to the Author):

Review of "Bimodal seismicity in the Himalaya controlled by fault friction and geometry"
by Dal Zilio et al.

This is a very interesting manuscript that tackles a problem of extreme societal importance: the earthquake potential of the Himalaya, and in particular the vicinity of the recent Gorkha, Nepal, earthquake. The authors use a new and innovative modeling method to simulate multiple seismic cycles (albeit in 2D) in a system that includes a tremendous number of realistic details, such as fault geometry, thermo-mechanical properties, and plastic failure. They experiment with a range of fault geometries and frictional properties to investigate which combinations lead to the observed bimodal pattern of seismicity in this area. The results may have important implications for our estimates of earthquake hazard - in particular, the authors' models imply that there may be plenty of stress left over in the Gorkha region for a much larger earthquake than the recent one, threatening Kathmandu.

Overall I think this is a well-designed study that should be published. I have some big questions about the method, though these questions probably betray my lack of familiarity with this type of modeling. It still is not entirely clear to me what constitutes a "fault" in the current method. I gather from the method description that rather than having slip on a surface, the earthquakes actually correspond to shear failure of a zone of finite width that fails in some sort of volumetric shear. The authors are able to define effective coefficients of friction, but they use static coefficients that are much smaller (<0.2) than are commonly used in "standard" spontaneous dynamic rupture models. Their S values are also quite high compared to those used in previous faulting studies. It also was not clear to me the time scale of the rupture events the authors are modeling. At one point the authors note that they have a time step of 1 year and that they have effectively a quasi-static formulation, but yet they make a point of their method including inertia. How this relates to the rupture and slip propagation documented in Figure S4 is not clear, because the times are scaled by the overall time of the rupture.

Yes exactly, the reviewer is right. In our continuum mechanics approach, faults are essentially predefined as a relatively weaker (i.e., with a lower static friction μ_s) thin layer. A simulated rupture event refers to the occurrence of rapid threshold-exceeding slip during which permanent displacement and stress drop occur along a localized interface. Plastic failure thus localises in a finite width through a volumetric shearing.

As pointed out by the referee, the static friction we assume for the MHT is low. However, such a low value refers to the effective friction, which already includes the pore-fluid pressure ($\lambda = 1 - P_{fluid}/P_{solid}$). The choice of such a low frictional value is based on a compilation of published data (please see next question).

With regard to the S parameter, our three reference models (Fig. 5) show higher S values with respect to those inferred from dynamic rupture simulations (e.g., *Gabriel et al., JGR, 2012*). However, this difference arises from the physics of our numerical experiments. Unlike the common dynamic rupture simulations, in which the fault is usually planar, the pre-stress is imposed on the fault, and the friction drop is sometimes very small (e.g., *Ma and Hirakawa, EPSL, 2013*), our models account for both complex fault geometry and self-consistent stress distribution along dip. This results in a large strength excess on the updip region of fault, which eventually leads to partial ruptures and a localised stress drop. However, as shown in Fig. 4 (panel C), an important parameter that controls the S parameter is the amount of slip velocity-induced weakening ($\gamma = 1 - \mu_d/\mu_s$). The choice of γ is based on laboratory experiments at coseismic slip rates (*Di Toro et al., Nature, 2011*). For the same static friction, the median S can significantly decrease when assuming a lower amount of slip velocity-induced weakening. We clarified this point in the manuscript at lines #196–201.

Our modelling approach is divided in two steps. During the first stage, we adopt a time step of 100 yr, thus allowing the stress to build up and the lithosphere to assume a setup in isostatic equilibrium, as well as to stabilise all the physical variables such as strain rate, temperature, viscosity etc. In the second modelling stage, the time step progressively decreases to 1 yr, while the inertia term and rate-dependent friction are activated. This means that our model produces unrealistically long seismic events because of the relatively large time step. However, the presented results generally demonstrate a satisfactory agreement with a wide range of long- and short-term natural observations of the Himalayan seismicity, as well as in term of slip and stress drop (as shown in *van Dinther et al., (2013, JGR)*).

In response to the concern raised by the referee, we have amended the Methods section by clarifying some parts and by including a section of “modelling procedure” and “modelling limitations”.

None of these questions are meant to be criticisms of the method---I am pleased that the researchers are applying a novel one, because it indeed looks like a reasonable way to model the seismic cycle. However, to be useful to a broader audience (in particular, one that is used to different earthquake modeling methods), the authors should take more care to lay out exactly how their method works, particularly during the dynamic rupture process, and argue more strongly for their choice of frictional parameters.

We understand the concern. The choice of frictional parameters is based on data gathered from several studies. The Himalaya is treated as a crustal-scale equivalent of an accretionary prism. This means that, given the average slope of the Himalayan wedge and the estimated dip angle of the MHT, and assuming an internal friction of 0.85, the effective basal friction on the flat portion of the MHT is inferred to be ~ 0.12 (*Avouac, 2015*, and references therein). This is in keeping with the numerical model of *Cattin and Avouac (2000)*, which was found to require a basal effective friction less than 0.13. This value is also consistent with the analysis of *Davis et al. (1983)* who considered the whole Himalayan wedge (which has a steeper slope than the Lesser Himalayan portion) and obtained a somewhat larger effective basal friction of 0.25. Moreover, The formal inversion of the thermochronological and thermobarometric data suggest an upper bound on shear heating induced by frictional sliding that, in turn, requires the effective friction coefficient to be ~ 0.07 (*Herman et al., 2010*).

In response to the concern raised by the referee, we have clarified the choice of the friction parameters in the Methods section.

I have a number of other, more minor comments, that I have included on annotated versions of manuscript and supplementary materials.

We went through the manuscript and tackled all minor comments one-by-one. Each correction are indicated in blue in the revised version of the manuscript.

I recommend the paper be published with minor revisions.

Reviewer #2 (Remarks to the Author):

This study uses a sophisticated thermo-mechanical model with elasto-visco-plastic rheology and velocity-weakening frictional plasticity to examine earthquake behavior on the MHT under the Himalaya. In particular, the authors focus this paper on the ? bimodal? behavior of earthquake ruptures in context of this model. Bimodal earthquake behavior is suspected to be the long-term behavior of the natural system based on historic and paleoseismic records of earthquakes.

This is indeed a very important problem in earthquake science and it is a fascinating setting with wonderful observations. The work in this paper is very impressive and this study makes an important contribution to the "debate" about bimodal seismicity and the geometry of the MHT. The natural evolution of bimodal events in the numerical simulations is clearly an important result.

I really like this study, and I am very impressed by the sophistication of the simulations and the intent to model earthquakes in a system where the stress and (to some extent) the thermal and rheological conditions evolve naturally. The authors have gone to great lengths to generate a mechanical system that is consistent with a wide range of geophysical observables and the model setting in Figure 2 is quite impressive -- the model is able to match first-order geodetic patterns as well as spatial distribution of off-MHT seismicity (although the seismicity is not shown, but it is described). It is however, very difficult to assess this model and digest all of Figure 2 because all of the methodology is buried in Supplementary materials and the Methods document. I understand this is the "nature" of Nature papers, but it does make it quite difficult to understand the technical aspects of this paper.

We agree. As pointed out by the first referee (please see above), and following the guide for submission, we have amended the Methods section (which will be at the end of the manuscript) by clarifying some parts and by including a section of "modelling procedure" and "modelling limitations". All changes are highlighted (in blue) in the revised version of the manuscript.

For this paper, the authors effectively treat Figure 2 and all the model development/ conditions to generate Figure 2 as the backdrop, or setting for the main point of the paper, which is how earthquakes naturally evolve in this mechanical system. I have no problem with this, but in my opinion, Figure 2 and the supplementary figures required to understand how Figure 2 could (perhaps should?) be a stand-alone paper itself. It is very impressive work. I emphasize that it is impossible to understand this Figure without working through the Methods and Supplementary sections, which again, is perhaps not a problem, but it requires the reader to work very hard.

The referee is correct; Figure 2 has been used to "validate" the model in reproducing appropriate interseismic stress and strain distributions which in turn result in a good match between the surface displacement and the real GPS stations. In response to the concern raised by both referees, the Methods section including the numerical approach, model setup, modelling procedure and limitations has been moved, in a single section, at the end of the manuscript. The Methods section will appear in the online research article and will contain all elements necessary for interpretation and replication of the results.

I do think it would be necessary to at least have a short paragraph in the main text describing the model setup which is currently entirely in the Supplementary information. In fact, Figures 2 and 3 are presented without any mention of friction parameters. It is necessary for the reader to know just a few basic conditions to understand Figures 2 and 3: friction parameters and spatial distribution (are the friction parameters the same on and off the MHT, for example?); basic boundary conditions, like how shortening is imposed; how is the MHT distinct from the bulk medium -- why does deformation localize on the MHT? It's not even clear whether these are dynamic simulations or quasi-static.

The referee is right that the model setup is not explained in the main text. We added a short paragraph at the end of the introduction in the main text describing the model setup, the kinematic boundary conditions, the three fault geometries adopted, and the range of effective static fault friction tested in the numerical experiments. A detailed description of the model setup is given in the Methods section.

One issue I am left wondering about after reading this paper is what, ultimately, is the physics controlling the rupture timing and spatial extent in this model? This is the entire point of the paper. Do you really need the sophisticated thermo-mechanical model to produce the bi-modal earthquake behavior, or does it all come down to the choice of friction parameter and friction drop, as illustrated in Figure 4? Would you get this bimodal behavior in, say, an elastic halfspace without any off-fault inelasticity? In the end, it seems the bimodal behavior is controlled entirely by the friction parameters and geometry. And from Figure 5 it appears that the bend in the fault is ultimately what is needed to get the truly bimodal distribution. I wonder if this bimodal behavior would arise from any model with some slip weakening and the mid-crustal ramp geometry?

Yes, this is the key point of this study. The bimodal seismicity arises from the relative distribution of pre-stress and fault strength. We show that the fault-bend can indeed result in appropriate distributions of pre-stress and strength. It would have been difficult to prove this point using a simpler model with no off-fault plasticity. Non-realistic stress concentrations would build up at the fault kinks if the medium was assumed purely elastic. In reality these stress concentrations are limited by the yielding strength of the medium as is the case in our model.

Some more specific questions/comments:

1. In the abstract lines 5-8. This sentence is awkward and confusing. You have a colon followed by a statement about the nature of bimodal seismicity. However the way this sentence reads to me, I would expect the phrase after the colon to be about the factors that regulate seismicity.

The entire abstract has been revised to improve the readability and to keep the text within ~150 words, as indicated in the journal guidelines.

2. Abstract lines 12-16. I know you have the word "may" in there, but these statements sound like you are saying that next rupture is likely to be much larger than the Gorkha, but your simulations suggest several Gorkha-size earthquakes can occur in sequence. I don't think you can say anything about the next earthquake? only that you can expect Gorkha-sized events more frequently than the M8+ events.

Yes, we agree, it might be misleading. We modified the abstract accordingly.

3. Line 36. You are using "mid-crustal" as a noun. Is that intentional? It looks like an adjective.

The word "ramp" was missing. We have amended the text to correct it accordingly.

4. Line 62. I don't think you ever explicitly stated that the EF geometry is shown in Figure 1b.

We agree. We have amended the text to clarify it.

5. Lines 80-84. It is unclear what you mean by "shortening on a non-planar dislocation"? You can't have shortening on a dislocation. Why do you have 19-20 mm/yr of convergence across the Himalaya when you impose 38 mm/yr of shortening according to the Methods section? Are you saying you impose 38 mm/yr of total shortening and 19-20 mm/yr occurs as slip across the MHT. Where does the rest of the shortening go? This needs to be cleaned up, I think.

We agree that this term is incorrect; we corrected it.

Yes, we imposed a convergence rate of 38 mm/yr, which results in total shortening of ~19-20 mm/yr. The remaining convergence rate vanishes through aseismic creep. We corrected the manuscript accordingly.

6. Figure 2 shows elastic, recoverable strain. How much strain is permanent, inelastic?
It would be interesting to see that, although I realize it isn't the purpose of this paper.

We followed the referee's advice and added a figure showing the inelastic deformation (Fig. S4).

7. In Figure 2, what component of strain is shown? I can't understand the colorbar label. Why is "interseismic strain" so localized on the MHT? If this fault is "locked" interseismically, I would expect the elastic strain to be quite diffuse. Is the fault zone more compliant (elastically) than the surrounding medium?

Figure 2 shows the elastic component of interseismic strain. We modified the text to clarify this point. Yes, the fault zone is indeed more compliant than the surrounding medium: The interseismic strain localises on the megathrust because the shear modulus is lower (see Table S1) and mostly because the strength is relatively lower than the surrounding medium due to a smaller static friction μ_s (please see second question from referee #1). We made this choice because, as described in numerical simulations incorporating damage rheology (e.g., *Lyakhovskiy et al. 1997; Lyakhovskiy et al., 2005*), there is an inverse relation between the degree of fault damage and the shear modulus. Further analysis on the interseismic coupling in our models indicate that the fault is highly coupled, but not fully locked (i.e., average interseismic coupling in the order of 0.90-0.95).

In response to the concern raised by the referee, we have clarified the choice of the frictional properties in the Methods section.

8. Line 90 is an awkward sentence. It doesn't make sense. What is "mechanical consistency"? Consistent with what?

The referee is correct that this definition was unclear. We have amended the text to clarify this point.

9. Line 103. I mentioned this previously, but you have not given any information about the conditions in the "reference model" (friction, etc.).

As stated above, we added a paragraph at the end of the introduction that describes the model setup.

10. Are friction parameters, μ_s and μ_d , uniform with depth on the fault?

The static friction (μ_s) is constant with depth, whereas μ_d depends on the local slip velocity-induced weakening. This means that, before a rupture event, the friction is constant throughout the fault. When an event nucleates and propagates in depth, the local slip velocity controls where, and to which extent, the dynamic friction (μ_d) decreases.

11. Why do you consider such very low friction coefficients? Coefficients less than 0.2 are considered quite low for rock. I believe some clay minerals and phyllosilicates have coefficients of friction as low as 0.2, but typically friction is much higher. Are you considering this as an effective coefficient of friction that includes pore pressure?

Yes, we consider the static friction as an effective friction that includes pore fluid pressure. These questions have already been tackled. Please see the first and second questions from referee #1.

REVIEWERS' COMMENTS:

Reviewer #1 (Remarks to the Author):

In the revised version of this manuscript the authors have done an admirable job of addressing the comments and questions of both myself and the other reviewer. I only have two small issues that could be quite easily addressed, I believe.

First, it would be helpful if the authors clarify that while their definition of pulse-like vs. crack-like rupture (depending on whether healing happens during rupture propagation or not) is consistent with the definitions used in the fault dynamics literature, the timescale is of course much longer than typical dynamic studies. Thus, it's not entirely clear that the physical causes or implications of pulse-like vs. crack-like rupture are the same in this model as in typical fault dynamics studies. Do the authors think that actual earthquakes (taking place on time scales of seconds to minutes) behave in the same way as their models, with the complete events tend to be crack-like, and the incomplete events tend to be pulse-like? The Gorkha event would seem to be evidence in that direction, but it would be helpful to make this point clearer.

In addition, it would be helpful to define exactly what is meant by the effective friction the first time it is mentioned, so that the reader will immediately understand why the value of μ -static is so low.

In reality, the paper can probably be published as-is in its revised form, but if possible I'd like to see those two issues addressed.

Reviewer #2 (Remarks to the Author):

I don't have a lot to say this time. My original review was very positive and I had only some minor concerns, mostly requiring clarification. The authors have done a nice job responding to these comments, as well as similarly minor comments from the other reviewer. The main challenge reading this manuscript is digesting all of the background numerical methods, and this was noted by both reviewers. The authors have addressed this by modifying the methods section and including some modeling details. Also, the authors also added a short paragraph in the main text to describe the model setup. I think this is an important improvement to the paper. I think the paper is easier to understand now.

As I said before, this is a very impressive study. The authors are doing important work to embed earthquake cycles Geodynamic models with evolving stress and deformation conditions. This work is quite far ahead of all other research groups working on such problems, in my view. This is an important paper for this reason, as well as for the specific results related to the Himalaya, which in itself is an important problem.

Bimodal seismicity in the Himalaya controlled by fault friction and geometry

Luca Dal Zilio, Ylona van Dinther, Taras Gerya and Jean-Philippe Avouac

Response to the Reviewer's Comments

October 28, 2018

>> Reviewers' comments:

Reviewer #1 (Remarks to the Author):

In the revised version of this manuscript the authors have done an admirable job of addressing the comments and questions of both myself and the other reviewer. I only have two small issues that could be quite easily addressed, I believe.

First, it would be helpful if the authors clarify that while their definition of pulse-like vs. crack-like rupture (depending on whether healing happens during rupture propagation or not) is consistent with the definitions used in the fault dynamics literature, the timescale is of course much longer than typical dynamic studies. Thus, it's not entirely clear that the physical causes or implications of pulse-like vs. crack-like rupture are the same in this model as in typical fault dynamics studies. Do the authors think that actual earthquakes (taking place on time scales of seconds to minutes) behave in the same way as their models, with the complete events tend to be crack-like, and the incomplete events tend to be pulse-like? The Gorkha event would seem to be evidence in that direction, but it would be helpful to make this point clearer.

We agree. We have amended the text to clarify it (line 139–159).

In addition, it would be helpful to define exactly what is meant by the effective friction the first time it is mentioned, so that the reader will immediately understand why the value of μ -static is so low.

We followed the referee's advice and we modified the text accordingly (line 68).

In reality, the paper can probably be published as-is in its revised form, but if possible I'd like to see those two issues addressed.

Reviewer #2 (Remarks to the Author):

I don't have a lot to say this time. My original review was very positive and I had only some minor concerns, mostly requiring clarification. The authors have done a nice job responding to these comments, as well as similarly minor comments from the other reviewer. The main challenge reading this manuscript is digesting all of the background numerical methods, and this was noted by both reviewers. The authors have addressed this by modifying the methods section and inducing some modeling details. Also, the authors also added a short paragraph in the main text to describe the model setup. I think this is an important improvement to the paper. I think the paper is easier to understand now.

As I said before, this is a very impressive study. The authors are doing important work to embed earthquake cycles Geodynamic models with evolving stress and deformation conditions. This work is quite far ahead of all other research groups working on such problems, in my view. This is an important paper for this reason, as well as for the specific results related to the Himalaya, which in itself is an important problem.

- - -

The authors acknowledge the insightful comments and suggestions from the anonymous reviewers and editors for improving the content of this manuscript.